# Quantification of Above-Ground Biomass over the Cross-River State, Nigeria, Using Sentinel-2 Data

Ushuki A. Amuyou [1,*], Yi Wang [1], Bisong Francis Ebuta [2], Chima J. Iheaturu [3] and Alexander S. Antonarakis [1]

1   Department of Geography, University of Sussex, Brighton BN1 9RH, UK
2   Department of Geography and Environmental Science, University of Calabar, Calabar 540271, Nigeria
3   Institute of Geography, University of Bern, 3012 Bern, Switzerland
*   Correspondence: ua53@sussex.ac.uk; Tel.: +44-774-101-8922

**Abstract:** Higher-resolution wall-to-wall carbon monitoring in tropical Africa across a range of woodland types is necessary in reducing uncertainty in the global carbon budget and improving accounting for Reducing Emissions from Deforestation and forest Degradation Plus (REDD+). This study uses Sentinel-2 multispectral imagery combined with climatic and edaphic variables to estimate the regional distribution of aboveground biomass (AGB) for the year 2020 over the Cross River State, a tropical forest region in Nigeria, using random forest (RF) machine learning. Forest inventory plots were collected over the whole state for training and testing of the RF algorithm, and spread over undisturbed and disturbed tropical forests, and woodlands in croplands and plantations. The maximum AGB plot was estimated to be 588 t/ha with an average of 121.98 t/ha across the entire Cross River State. AGB estimated using random forest yielded an $R^2$ of 0.88, RMSE of 40.9 t/ha, a relRMSE of 30%, bias of +7.5 t/ha and a total woody regional AGB of 0.246 Pg for the Cross River State. These results compare favorably to previous tropical AGB products; with total AGB of 0.290, 0.253, 0.330 and 0.124 Pg, relRMSE of 49.69, 57.09, 24.06 and 56.24% and −41, −48, −17 and −50 t/ha bias over the Cross River State for the Saatchi, Baccini, Avitabile and ESA CCI maps, respectively. These are all compared to the current REDD+ estimate of total AGB over the Cross River State of 0.268 Pg. This study shows that obtaining independent reference plot datasets, from a variety of woodland cover types, can reduce uncertainties in local to regional AGB estimation compared with those products which have limited tropical African and Nigerian woodland reference plots. Though REDD+ biomass in the region is relatively larger than the estimates of this study, REDD+ provided only regional biomass rather than pixel-based biomass and used estimated tree height rather than the actual tree height measurement in the field. These may cast doubt on the accuracy of the estimated biomass by REDD+. These give the biomass map of this current study a comparative advantage over others. The 20 m wall-to-wall biomass map of this study could be used as a baseline for REDD+ monitoring, evaluation, and reporting for equitable distribution of payment for carbon protection benefits and its management.

**Keywords:** above ground biomass (AGB); REDD+; Nigeria; Sentinel-2; random forest

## 1. Introduction

Tropical forests encompassing less than a fifth of the Earth's terrestrial area [1] are one of the most important components of global terrestrial ecosystems, accounting for around 55% of total aboveground biomass (AGB) [2–4], hold two-thirds of global biodiversity [5,6], sustain the economy of millions of rural populations and contribute to climate regulation [7]. However, a recent analysis revealed that the tropics are now a net carbon source rather than a carbon sink, attributed mainly to anthropogenic land cover changes [8–10]. In addition, changes in climate patterns and variability will also begin to have a serious impact on tropical forested landscapes, especially those of Africa [11].

African land cover encompasses diverse types of woody and forested landscapes as well as a patchwork of undisturbed and disturbed forests, and wood species present within heterogeneous farmed lands [12]. These diverse land cover types have varied AGB densities even within a landcover type [13]. For instance, Saugier et al. [14], Knelling et al. [15], IPCC [16] and Gibbs et al. [17], estimated mean AGB of 390 mg/ha, 190 mg/ha, 400 mg/ha and 198 mg/ha, respectively, in African intact forests. Bouveta et al. [13] summed the varied estimates of AGB in Africa and concluded that the region's savanna and woodlands contained 52% of the total AGB while intact forests contained 48% of the AGB. One of the reasons for this variation is that tropical forests do not have a universally agreed definition, and in Africa, there are a variety of tropical landscapes from wooded savannas, to humid tropical, to closed tropical and dry tropical forests [18]. In effect, nearly 75% of Africa's forests are considered woodland savannas and dryland forests [19], with carbon storage in African tropical forests only accounting for around 48% of the total [20]. Another reason is the paucity of forest inventory plots available in Africa to estimate AGB and calibrate/validate remote sensing derived biomass products [21], compared to other tropical regions.

Different tropical-wide AGB maps have been produced in the last decade using a combination of satellite data and ground-based plots. Saatchi et al. [22] first produced a biomass map using satellite LiDAR and MODIS data and a machine learning spatial extrapolation method at a fine resolution of 1 km. They used 75 calibration forest inventory plots over Africa, producing a total AGB estimate of 124 Pg with an uncertainty of ±32%. Baccini et al. [23] also used satellite LiDAR and MODIS data within the random forest framework to predict AGB over the tropics at a 500 m resolution. They calibrated their product using 283 plots throughout the tropics, producing a total Africa AGB estimate of 129 Pg and an average RMSE of 38 t/ha. In a more recent study, Avitabile et al. [24], fused the Saatchi and Baccini products at 1 km, producing AGB over Africa of 96 Pg with an RMSE of 83.7 t/ha (a reported improvement of around RMSE 20–30 t/ha compared to the Saatchi and Baccini products). Furthermore, they used 953 reference points over Africa out of over 14,000 in the tropics. Therefore, the Saatchi et al. [22], Baccini et al. [23] and Avitabile et al. [24] studies produced their products with limited calibration and validation plots in Africa. Similarly, Santoro and Cartus [25], henceforth referred to as the European Space Agency Climate Change Initiative (ESA CCI) Biomass project, estimated total tropical AGB to be 331.3 Pg and Africa having AGB stocks of 84.4 Pg. These varied estimates over the same region from different authors derived from various remote sensing instruments and protocols with little timelapse contributions to the high AGB uncertainty and lack of effective carbon stock tracking and management.

Article 2.1 of the Kyoto Protocol highlighted the need for individual countries to reduce GHGs to 'a level that would prevent dangerous anthropogenic interference with the climate system' [26,27]. The articulation of the Kyoto Protocol was the seed that led to the formation of the Reducing Emissions from Deforestation and Degradation (REDD) [28]. REDD, created by the UNFCCC Conference of Parties, encourages countries to contribute to climate change mitigation through reducing emissions from deforestation and forest degradation, and increasing the removal of greenhouse gases (GHGs) through sustainable management of forests and the conservation and enhancement of forest carbon stocks. To attain this goal, developed countries were encouraged to focus on fossil fuel related emissions while tropical developing economies were commissioned to concentrate on LULCCs linked emissions especially from the forestry sector [29,30]. More so, tropical country AGB quantification supports the monitoring of biodiversity status [31], protects carbon pools [32] and increases social and environmental ecosystem services to forest communities who largely depend on natural resources for daily subsistence [33].

In addition, studies determining AGB density in Nigeria, and specifically tropical Nigeria, have not used local forest inventory plots to calibrate biomass estimation [22–24,34]. In these studies, reference points from the Republic of Congo, Uganda, Ghana, Cameroon, etc. were used for model calibration and results were extrapolated to Nigeria without

any point dataset collected from there despite the differences in vegetation disturbance history, plant functional types, soils and climate which affect biomass density [34]. In addition, the IPCC biomass estimation guide [35] advised that for better biomass estimation accuracy, tier three level (which is country or subnational) estimation of biomass should be encouraged. These sub country regional biomass estimations can then be agglomerated to obtain national biomass density and spatial variations for effective verification, reporting, monitoring (MRV) and subsequent payments of subventions under the REDD+ initiative.

Cross River State has more than 50% of Nigeria's remaining tropical intact forest and is one of the 25 biodiversity hotspots of the world [36]. However, the ecological integrity of the region is under threat from anthropogenic destruction [37]. In 2020, Global Forest Watch [38] revealed the state had lost 12.7 Kha of its tree cover. The rate of land-cover change (at 3.7%) in Nigeria per year remains among the highest in the world [36]. The destruction of tracks of forest cover leads to biomass loss, but REDD+ in 2018 [39] estimated 0.267 Pg of above ground biomass in the state. The REDD+ project in Cross River State did not carry out wall-to-wall AGB estimation and the field campaign was restricted to tropical forested zones. Other land cover types, such as disturbed forests, mixed agroforest areas and savanna landscapes, were left out of the UN-REDD+ Nigeria study [39]. Tree heights were not measured in the field but were derived using the Feldspausch et al. [40] height-diameter tropical forest allometry.

In these contexts, the aim of the study is to derive high spatial resolution (20 m) AGB for the whole of the Cross River State, Nigeria, using Sentinel-2 data, climatic and edaphic variables and with local reference forest inventory plots taken from undisturbed, disturbed and cropland areas. We use Sentinel-2 data and forest inventory plots collected concurrently in 2020 to produce a regional AGB map. Specifically, the study planned to (1) establish a network of forest inventory plots in a variety of forest and woodland landscapes for AGB estimation and (2) use Sentinel-2, climate and soil variables to predict and spatially extrapolate AGB to the Cross River State using random forest machine learning, and (3) compare the AGB map of this study with well-known products from Baccini, Saatchi, Avitabile and ESA CCI as well as comparing to the REDD+ AGB estimates published regarding the Cross River State.

## 2. Materials and Methods

### 2.1. Study Area

The study area is the Cross River State in southeast Nigeria, with an area of 20,156 km$^2$ (Figure 1). The area covers an elevation range from 1800 m in the extreme north to 103 m above sea level in the southern part of the state [39]. It shares boundaries with Benue State in the north, Akwa Ibom, Ebonyi and Abia States in the west and the Atlantic Ocean to the south. Cross River State has five different vegetation types: mangrove, swamp and tropical rainforest which dominate the southern and central parts of the region, montane vegetation and savanna woodlands are dominant in the northern portion of the study area [37]. It is recognized as one of the biological hotspots in the world [41] and two locations—Oban and Okwangwo—are marked out as conservation spots. The Oban Division (OD) covers an area of 2800 km$^2$ with 1568 identified plant species while the Okwangwo Division (OkD) has a land area of 800 km$^2$ with 1545 plant species located in the area [42]. Analysis of the extent of land cover types in the region shows that mangroves occupy 480 km$^2$, swamps 520 km$^2$, tropical rainforest 7290 km$^2$, plantations 460 km$^2$, other forests 216 km$^2$ and other land uses 12,300 km$^2$ [43].

Rainfall in the Cross River State is bimodal with differences across the three agroecological zones (AEZs). The rainfall gradient is largely influenced by relief and nearness to the coast. The southern AEZ has a tropical monsoon climate with an annual mean rainfall of 3500 mm, which sometimes peaks at 4000 mm around the Oban Massif [44]. The climate of the region is within the tropical monsoon (Am) classification scheme of Koppen [45]. The mean annual air temperature of the zone averages around 27 °C with little variation throughout the year, and with humidity between 78% and 91% [46]. In the central AEZ,

the mean annual rainfall varies from 2300 to 3000 mm. The zone records mean annual air temperature ranging from 26.9 to 30 °C and the humidity of the zone in most parts of the year is about 68% [44]. In the northern AEZ, the savanna ecosystem is common with a mean annual rainfall of 1120 mm and air temperature ranging from 15 to 30 °C [47]. The zone has two climate seasons: the rainy season, which lasts for about eight months, and the harmattan, which lasts for about four months. In the montane ecoregion of the Obanliku Mountains within the northern AEZ, climatic conditions are markedly different from other parts of the region. Air temperature have a mean annual range of 4–10 °C. The terrain is rugged with hilly escarpments, steep valleys and mountains that peak at about 1800 km² above sea level with an elongation into the southwest region of Cameroons [44].

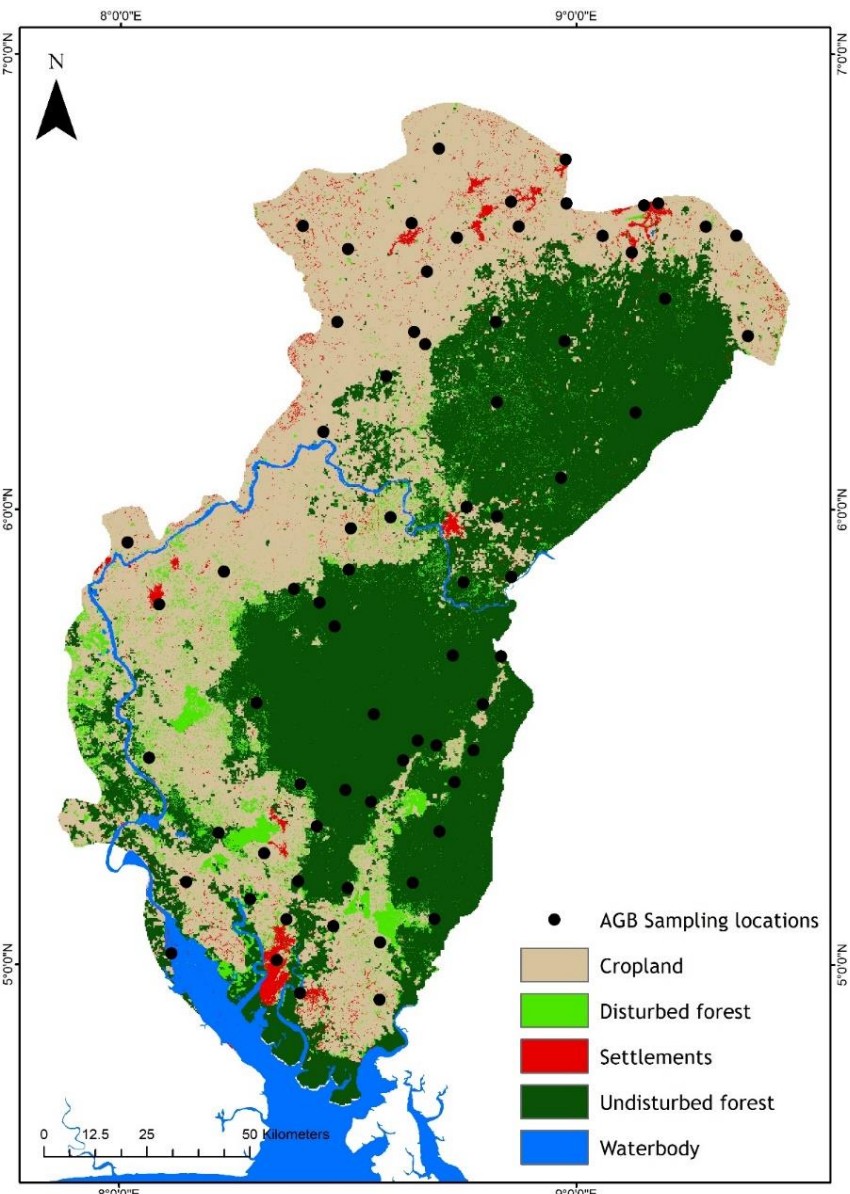

**Figure 1.** Forest inventory plots throughout the Cross River State were established with Forestry Commission guidance following their land cover classification (Cross River State Forestry Commission Forestry Manual 2019 [48]).

*2.2. Forest Inventory Survey*

A land cover map (Figure 1) developed by the Cross River State Forestry Commission [48] was used in establishing the forest inventory plots. The study area was classified

into: undisturbed forest (UF), disturbed forest (DF) and croplands (CF) based on the Cross River State Forestry Commission staff guide and with modification of Gautam and Mandal delineation [49]. The undisturbed land covers considered in this study were unbroken stretches of land covered with diverse tree species with little or no human interference in the ecological structure while those with evidence of anthropogenic activities, such as tree stumps and patches of logging, roads, pronounced footpaths, banana and cocoa farmland patches, farm huts and any gap in the forest land, were attributed to human activities [50]. On the other hand, croplands or agroforestry areas are woodlands with different species of crops cultivated in them at the same time. Photographs providing examples of measured plots are given in the Supplementary Materials S1.

The GPS points of purposively chosen locations were overlaid on a map of the Cross River State across the landcover types identified for this study. Thereafter, GPS coordinates of each chosen sample point were inserted into the GPS Garmin eTrex model (with accuracy of 3 m), and on the ground, the Goto function was used to locate the plot for the inventory. In all sample locations, entry point was through a known community [39]. Accordingly, 29 plots were established in undisturbed landcover, 18 in disturbed land cover and 25 were established in croplands. It should be noted that chosen plots that were difficult to assess on account of geomorphic features, such as river, flooded streams, steep slopes or security challenges, such as intercommunity or interstate clashes, resulted in other alternative locations being chosen.

The field campaign commenced in March 2020 and ended in November of the same year. In this study, 72 nested square plots of 20 m × 20 m were established. Trees of sizes >50 cm, 20–50 cm and <20 cm diameter at breast height (1.3 m) were inventoried in the 20 m × 20 m plots and subplots of 15 m × 15 m and 7 m × 7 m, respectively [50]. In each of the 72 plots, all tree species were identified, numbered and DBH measured using a measuring tape and the total height was taken with a Trupulse Criterion RD 1000. Given that the wood density of tropical trees species is erratic [51], the study extracted wood density of each tree species identified from the African Wood Density Database provided by the World Agro-forestry Centre [52,53] and the African Wood Density of the Food and Agricultural Organization [54]. However, where the tree species wood density was not found in either of these databases, the mean wood density of the plot was used as the wood density of the tree species [50].

The allometric equation of Chave et al. [53] was used to estimate the AGB of each tree in each forest inventory plot. Chave's allometric equation requires total height, *H* (m), species wood density, $\rho$ (g/cm$^{-3}$), and diameter at breast height, *DBH* (cm), to estimate tree-level AGB. Chave et al.'s AGB estimation equation is given as:

$$AGB_{est.} \text{ (kg)} = 0.0673 \times (\rho \times DBH^2 \times H)^{0.976} \qquad (1)$$

The biomass of each tree within a plot was summed to obtain the total biomass per 400 m$^2$ plot in kilograms (kg/m$^2$) [55]. This is converted to tons per hectare. Figure 2 provides a synopsis of the dataset sources, analytical procedures and final AGB product of the study.

### 2.3. Regional Aboveground Biomass Estimation

This section presents the relevant spatial variables used in predicting regional aboveground biomass in the Cross River State, Nigeria, from the different sources and techniques used in the acquisition of Sentinel-2 vegetation indices, mean air temperature and the mean rainfall data over the study area.

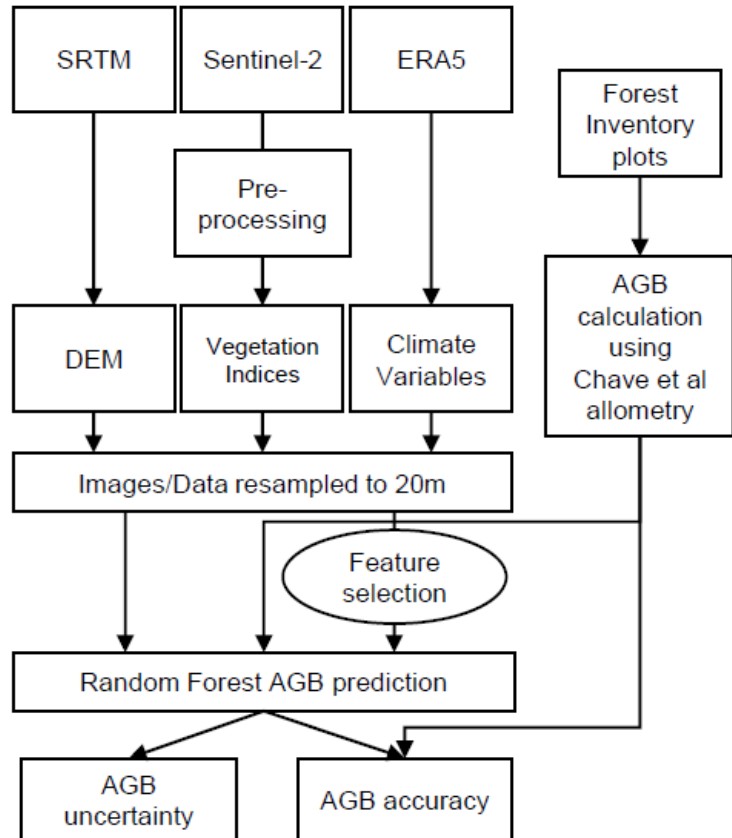

**Figure 2.** Methodological workflow showing data sources, analytical procedures, final output and accuracy assessment. Climate variables used were air temperature, precipitation, relative humidity and soil moisture. Vegetation indices are given in Section 2.3.1 below.

### 2.3.1. Satellite, Climatic and Topographic Variables

In this study, we utilized a total of eight Sentinel-2A multispectral images (hereafter called S2) alongside climatic and topographic variables. The S2 data were downloaded from the United States Geological Survey (USGS) Earth Explorer site at: https://earthexplorer.usgs.gov/ (accessed on 20 November 2020). The downloaded S2 level-1C (LIC) images were then transformed from radiance to surface reflectance aided by the dark object subtraction (DOS) method based on the semi-automated classification plugin in QGIS version 2.14 software [56]. With this process, all the darkest pixels caused by atmospheric scattering that may reduce the image quality are reduced [14]. The S2 images were atmospherically modified, orthorectified and spatialized on the global reference system UTM/WGS 84, 32N Minna datum on the SEN2COR tools of SNAP (Sentinel Application Platform) toolbox of the European Space Agency. Top-of-atmosphere (TOA) reflectance was converted to top-of-canopy (TOC) reflectance [57,58]. Sub-setting and mosaicking were carried out to produce a single image for the study area [59]. The S2 MSI (10 m) images were resampled to 20 m resolution to match the plot size (20 m) and this was done using the nearest neighbored resampling technique in ArcMap [59]. This interpolation method was used because its processes are faster, the algorithm has less rigorous implementation procedures and it is suitable for discrete data such as AGB [60–63]. The benchmarked image was then subjected to a geometric preprocessing protocol. All the images were downloaded from the last month (November 2020) of the field data campaign where weather conditions over the region are often less cloudy.

The various signal bands and vegetation indices were considered in this study and are shown in Table 1. The vegetation indices include the Normalized Difference Vegetation Index (NDVI), Enhanced Vegetation Index 2 (EVI 2), Optimized Soil Adjusted Vegetation Index (OSAVI), Modified Soil Adjusted Vegetation Index (MSAVI), Atmospherically Re-

sistant Vegetation Index (ARVI), Inverted Red-Edge Chlorophyll Index (IRECI), Modified Red-Edge Normalized Difference Vegetation Index (MRENDVI), Modified Red-Edge Simple Ratio (MRESR) and Red-Edge Normalized Difference Vegetation Index (RENDVI). In each of the delineated plots, the spectral reflectance values at the center point of the plot were extracted using the 'Extract Values to Points' spatial analytical tool in ArcGIS. This tool extracts the cell values of the raster dataset based on the plots (point features taken at the center of the plot). The equations used to calculate the above vegetation indices and their references are shown in Table 1. These vegetation indices were used because previous studies [22,24,25,34] established that these VIs are sensitive to phenological dynamics in vegetation, hence they can be used as proxies of forest biomass. More so, 30 m elevation data from the Shuttle Radar Topography Mission (SRTM) was downloaded from the United State Geological Survey's Earth Explorer (https://earthexplorer.usgs.gov/ (accessed on 20 November 2020) and subsequently resampled to 20 m spatial resolution using ArcMap's nearest neighborhood method [58].

Thirty-five years (1985–2020) mean annual air temperature, precipitation, relative humidity and soil moisture data over the Cross River State, Nigeria, were obtained from the European Centre for Medium-Range Weather Forecasts (ECMWF) ERA5 dataset, downloaded from the Copernicus Climate Change Service (S3C) Climate Data Store (https://cds.climate.copernicus.eu/ (accessed on 30 December 2020) [64]. The ERA5 are 5th generation elite model-based data produced on ECMWF Integrated Forecasting System (IFS). The ERA5 merged model-derived data with historical in situ and space-borne observational data under a robust quality control protocol. The ERA5 data are presented with a resolution of around 30 km. Subsequently, these climate parameters were upscaled to 20 m spatial resolution using the nearest input grid points as provided by Digital Earth Africa user guide (https://docs.digitalearthafrica.org/en/latest (accessed on 12 March 2021).

**Table 1.** Vegetation indices calculated from Sentinel-2 used in the study. Blue, Red, RE1, RE2 and NIR correspond to the Sentinel-2 bands 2,4,5,6 and 8.

| Vegetation Indices | Equations | References |
|:---:|:---:|:---:|
| NDVI | $(NIR - Red)/(NIR + Red)$ | [65] |
| EVI | $2.5 \times ((NIR - Red)/(1 + NIR + 6Red - 7.5Blue)$ | [66] |
| OSAVI | $(NIR\text{-}Red)/(NIR + Red + 0.16)$ | [67,68] |
| MSAVI | $(2 \times NIR + 1 - sqrt[(2 \times NIR + 1^2 - 8 \times (NIR - Red)])/2$ | [69] |
| ARVI | $(NIR - (2Red - Blue))/(NIR + (2Red - Blue))$ | [70] |
| IRECI | $(NIR - R)/(RE1/RE2)$ | [71] |
| MRENDVI | $(RE2 - RE1)/(RE2 + RE1 - 2 \times Blue)$ | [72] |
| RENDVI | $(RE2 - RE1)/(RE2 + RE1)$ | [73,74] |
| MRESR | $(RE2 - Blue)/(RE1 - Blue)$ | [75] |

### 2.3.2. Regional AGB Estimation Using Random Forest

Estimation of AGB across the Cross River State was based on Breiman's [76] random forest (RF) model. The RF is an ensemble decision tree algorithm used in both classification and regression analysis [77]. In regression analysis, the algorithm builds a series of decision trees on bootstrap samples and then takes the average of the output of each tree. The averaging reduces the variance of the model and improves its prediction accuracy. The accuracy of the prediction increases with an increasing number of trees [78–80]. The inherent ease of manipulation, the capacity to be executed with small sample sizes [81,82] and most importantly, overcoming overfitting and collinearity of variables challenges associated with complex data domains [76,83,84], make this method very appropriate in determining aboveground biomass [23,85,86] in this study. RF has two important features: Ntree and Mtry. Ntree is the number of decision trees formed based on the bootstrap samples of the observation which by default is 500, while Mtry is the number of variables used as potential candidates at each split [84]. Furthermore, to optimize model

performance, given the field samples and input layers, Ntree and Mtry were tested in the ranges of 250–1000 and 1–16, respectively. The optimal combination of Ntree and Mtry for AGB prediction was 400 and 3, respectively. The Ntree and Mtry used were enough to stabilize the error as too many Ntree may over correlate the ensemble and subsequently lead to over overfitting [79].

Concerning training and testing, 70% of the data (in bag sample) were used to train the model while the remaining 30% of the data (out-of-bag sample-OOB) were used for the internal cross-validation procedure for estimating the OOB error [80,86]. The $R^2$, RMSE and relative RMSE (relRMSE) of the model were used to interpret the relationship between the field obtained AGB and predicted AGB [87,88]. The relRMSE is defined in this study as the RMSE divided by the mean of the observed values. In addition, the selection of important features becomes crucial because of the interconnectedness and high dimensional properties of biophysical parameters [87,89]. Feature selection in random forest can be conducted using the filter, wrap or embedded method [90,91].

In this study, the recursive feature selection wrap method was used [92]. This method searches for the best subset of variables by adding (forward selection), eliminating (backward selection) or searching for the optimal subsets of variables (recursive selection) and ordering them based on their performance. This aided us in the reduction of the computational time, improvement in model performance with the right subset combinations, reducing overfitting and increasing the ease of data interpretation, among others [93]. The random forest algorithm has an inbuilt capacity to calculate the contribution of each of the explanatory variables to the model. The increased percentage in mean square error (%inMSE), computed as the prediction error of each tree on the out-of-bag samples as the data are randomly shuffled [76], is one measure that revealed the contribution of a variable to the model. Variables with higher values are indicative of their robustness in the model [94]. Node impurities tell us how well the variables split. It expresses the total decrease in impurities as the variables are divided during permutation and averaged over all the trees. In other words, it is the residual sum of squares as the features are divided [76]. MSE and node purities in random forest algorithms are the most widely used variable scores of importance in ecological studies [65,78]. The model parameter optimization process of the RF model is provided in the Supplementary Material.

To evaluate the effectiveness of the random forest model, the coefficient of determination ($R^2$), root mean square error (RMSE) and the percentage mean square error (i.e., relRMSE) were used to determine the general error of the AGB estimation. Generally, a high $R^2$, with low RMSE and relRMSE, is an indication of a good predictive model [70].

$$RMSE = \sqrt{\frac{\sum_{i=1}^{n}(y_i - \hat{y}_i)^2}{n}} \tag{2}$$

$$relRMSE\% = 100 \cdot \left(\frac{RMSE}{\hat{Y}}\right) \tag{3}$$

where $y_i$ is the predicted value series, $\hat{y}_i$ is the observed value series, $n$ is the sample size and $\overline{Y}$ is the average value of the observed series. In addition, the field plot data were compared with the extracted AGB values from Saatchi, Baccini, Avitabile and ESA CCI AGB maps using the Willmott's agreement index, as shown in Equation (4).

$$d = 1 - \frac{\sum_{i=1}^{n}(O_i - P_i)^2}{\sum_{i=1}^{n}\left(|P_i - \overline{O}| + |O_i - \overline{O}|\right)^2} , \; 0 \leq d \leq 1 \tag{4}$$

where $O_i$ is the AGB from field plots, $\overline{O}$ is the observed mean AGB and $P_i$ is the AGB values from each of the maps used in this study [95]. An index of 1 implies a perfect agreement between a pair of datasets. The Willmott index (d) is a standardized statistical technique used to establish the extent of prediction error which varies between 0 and 1 [95]. Willmott [95] reported that the index of similarity is not sensitive to errors concentrated



around outliers. In addition, it is simple to implement and dimensionless, hence, the unit of data collection does not count. The Willmott index was used to support traditional model evaluation measures of $R^2$, RMSE and bias [96,97].

Errors in the AGB estimation could filter in at any stage of the research process: plot design, data collection, model formulation and parameterization or analysis [18]. To create the AGB uncertainty we assumed that the identified error sources are independent and random, and we propagated these errors to the pixel level using the formula [23]:

$$\varepsilon_{AGB} = \left( \varepsilon_{measurement}^2 + \varepsilon_{allometry}^2 + \varepsilon_{sampling}^2 + \varepsilon_{prediction}^2 \right)^{1/2} \tag{5}$$

This study uses Chave's et al. [53] pan tropical allometric equation to estimate the AGB and is associated with an error margin of 5%. The measurement errors of wood density, tree height and diameter at breast height in the region are estimated to be 10, 2.5 and 4.47%, respectively [22]. Similarly, the sampling error was taken from Saatchi [22] to be 22.8% for the tropics.

RF is a non-parametric ensemble technique, which does not require direct quantification of prediction error such as traditional regression approaches [78]. We therefore rely on the Monte Carlo model in quantifying the prediction uncertainty. The underlying principle of the Monte Carlo model is the repeated simulation of the occurrence of a random event and the subsequent estimation of its probability features based on the frequency of the said random event [98]. With repeated simulations of the Monte Carlo samples (in our case, 500 iterations), the probability distribution of biomass estimates, and errors are obtained from the series of iterations which resulted in a stable and reliable quantification of biomass and the error map [99].

These diverse error sources are propagated during the geospatial modelling process assuming all errors were independent and random; hence it is imperative to know their size and the pattern of distribution in accordance with IPCC and the Carbon Fund Methodological Framework [39].

*2.4. Comparisons to Other Regional to Global AGB Products*

A few tropical and global remote-sensing-based AGB maps have been produced in the past decade. In this study we will compare our AGB product over the Cross River State with that of Saatchi et al. [22], Baccini et al. [23], Avitabile et al. [25] and the ESA CCI [25]. These studies are summarized below. Furthermore, details of the Nigeria UNREDD+ project which has estimated AGB for the whole CRS in 2018 are also provided.

*Saatchi*: Saatchi integrated plot based AGB and GLAS (Geosciences Laser Altimeter System) LiDAR Lorey heights derived AGB with MODIS (NDVI and Leaf Area Index), QSCAT (NDVI and LAI) and SRTM (topography) to extrapolate AGB over the tropics at 1 km spatial resolution using the Maxent machine learning tool. Saatchi used 75 plots of 0.1 ha in size (493 in all tropics) scattered across tropical African forests, wood savanna and dry forests of Cameroon, Uganda, Liberia and Gabon and inventoried trees with DBH of 10 cm and above. Saatchi used an allometric equation that included tree DBH and wood density in estimating the AGB plot. The model predicted the total AGB for Africa to be 62 Pg. In addition, 40% of the point dataset were reserved for model testing while field plot datasets were bootstrapped and used with GLAS LiDAR to account for pixel per pixel error through the Maxent model. Saatchi examined the model performance based on two parameters: the segment of predicted area and extrinsic omission rate at a selected threshold and the area under the receiver curve (AUC). The Maxent model revealed that the AUC ranged between 0.86 and 0.98, indicating that the prediction did not happen by chance. The overall uncertainty averaged over all continents was also reported to be between ±30% and ±32% over Africa.

*Baccini:* Baccini determined a pan-tropical map using similar methods to Saatchi, but with the use of RF. Baccini measured all trees with DBH of 5 cm and above and produced the AGB map at 500 m spatial resolution. The AGB over the study area was

predicted and mapped using a random forest learning algorithm. Baccini used an allometric equation that includes tree DBH, height and wood density in estimating plot level AGB and 10% of the data were used to test the RF model. Additional spatial layers used as input data included surface temperature from MODIS bands, EVI2, NDVI2 and all land bands. Baccini produced a total AGB of tropical Africa at 64.5 Pg. Validation using their testing dataset resulted in an RMSE of around 50 t/ha for all tropical regions, with 38 t/ha for tropical Africa.

*Avitabile:* The AGB products of Saatchi and Baccini were combined into a pan tropical AGB map at 1 km resolution, using an independent reference dataset of field observations and locally calibrated high-resolution biomass maps. The data fusion approach used bias removal and weighted linear averaging incorporating the biomass patterns indicated by the reference data. Avitabile screened and selected 14,477-point data across the tropics and 953 were taken from Africa (DRC, Tanzania, Ghana, Ethiopia, Sierra Leone). Trees with a DBH range of 5–10 cm were used in model calibration and subsequently estimated 84 Pg as the total carbon stocks over Africa. The plots and GLAS LiDAR-derived AGB were spatialized using a random forest model. This fused product compared to the Saatchi and Baccini product, using its own validation dataset, reported RMSEs of 89, 104 and 112 t/ha and bias of 5, 21 and 28 t/ha, respectively.

*ESA CCI:* Here, the authors estimated growing stock volume (GSV) obtained mainly from radar data with a spatial resolution of 1 km. The GSV was converted to AGB using wood density and a stem-to-total biomass expansion factor. A total of 110,897 plots scattered across the globe were used in model validation. ESA CCI-derived AGB was integrated with CCI Land Cover datasets and using the Forest Resources Assessment (FRA) ecological zones of 2010. The ESA CCI estimated total AGB of 84.8 Pg for Africa against FRA estimates of 95.5 Pg with a mean AGB of 108 t/ha and 142 t/ha, respectively. The large variance between the two studies was attributed to the use of more forest area in the ESA CCI studies compared to FRA. AGB was predicted with a standard deviation around 50% for tropical forests and tropical mountain forests. RMSEs were provided with a range of AGB values, giving RMSEs of 30–50 t/ha for AGB > 100 t/ha and 50–100 t/ha for AGB < 100 t/ha.

*Nigeria UNREDD+ project:* Nigeria secured approval for the REDD+ project implementation in 2010 with Cross River State as a demonstration model. Cross River State holds 50% of the remaining 9.6 million hectares of Nigeria's forest area but is under threat of deforestation [39]. In addition, the region was selected for the first REDD+ implementation in the country based on the streams of forest governance structures and its carbon sequestration potentials [28]. The project established 77 nested plots of 35 m × 35 m across 13 land cover types for tree parameters inventory. Tree DBH was measured in the field while height and wood density were derived from the equations of Feldpauch et al. [40] and Zanne et al. [100]. The Chave et al. [18] allometric equation was used to estimate tree AGB. Tree AGB was summed to obtain plot-based AGB. Using a biomass conversion factor of 0.47, the estimated AGB of the region was given as 2544 t/ha.

*Extracting AGB from the regional products:* The four AGB products of Saatchi, Baccini, Avitabile and the ESA CCI were evaluated against the 22 testing forest inventory plots collected as part of this study. The Saatchi, Baccini, Avitabile and ESA CCI products were downloaded, the study area cropped, and projection parameters selected to conform with the coordinate system of the study area (UTM/WGS 84, 32N Minna datum) using the SEN2COR tools of the SNAP (Sentinel Application Platform) toolbox of the European Space Agency. To ensure effective comparison, each of the products' native resolution was used [25]. The forest inventory plots of this study were then overlaid on the subset AGB maps of Saatchi, Baccini, Avitabile and ESA CCI and compared to the extracted AGB values from each product.

## 3. Results

### 3.1. Summary Analysis of Plots AGB

Descriptive characteristics of the forest stands' features are presented in Table 2. Overall, there were 28 plots collected in undisturbed forests (Figure 1), 18 plots collected in the disturbed forests and 26 plots collected in crop field plots (henceforth UF, DF and CF). The mean number of trees in the UF, DF and CF plots were 38.8, 4.02 and 25.2 cm, respectively, while the mean heights of trees in these three land cover types were 23.6, 22.0 and 8.2 m, respectively. Basal areas on average were 35.5, 28.8 and 15.9 m$^2$/ha and average AGB was 222.5, 106.5 and 24.4 t/ha in the UF, DF and CF plots, respectively. Specific wood density (g/cm$^3$) ranged from 0.20 to 0.93 across all sites, with average wood densities of 0.71, 0.55 and 0.50 g/cm$^3$ in UF, DF and CF plots, respectively.

**Table 2.** Descriptive statistics of forest inventory plots.

| Landcover Type Parameters | Undisturbed Forest (*n* = 28) | | | Disturbed Forest (*n* = 18) | | | Crop Fields (*n* = 26) | | |
|---|---|---|---|---|---|---|---|---|---|
| | Max | Min | Mean | Max | Min | Mean | Max | Min | Mean |
| DBH (cm) | 164.4 | 5.1 | 38.8 | 164 | 5.1 | 40.2 | 82.6 | 5.1 | 25.2 |
| Tree height (cm) | 67.0 | 2.8 | 23.6 | 45 | 4.1 | 22.0 | 30.0 | 1.5 | 8.2 |
| BA (m$^2$/ha) | 77.4 | 6.3 | 35.5 | 105.4 | 5.9 | 28.8 | 43.6 | 2.7 | 15.9 |
| WD (g/cm$^3$) | 0.51 | 0.23 | 0.71 | 0.93 | 0.20 | 0.55 | 0.87 | 0.23 | 0.50 |
| AGB (t/ha) | 588.3 | 11.5 | 222.5 | 203.3 | 14.4 | 106.5 | 107.3 | 3.0 | 24.4 |

### 3.2. Predicting AGB Using Random Forest Algorithm

The result of the random forest training performance using all the explanatory variables (*n* = 16) results in a coefficient of determination of 0.85, an RMSE of 28.71 t/ha and MAE of 30.02 t/ha. As stated in the methodology, performing feature elimination is an important step in reducing the effects of multicollinearity and overfitting. The random forest algorithm has an inbuilt capacity to calculate the contribution of each of the explanatory variables to the entire model through the variable importance measures (VIMs). This is achieved using IncMSE and IncNodePurity (Figure 3). The MSE and node purity are filters used to rank and removed irrelevant variables from the model.

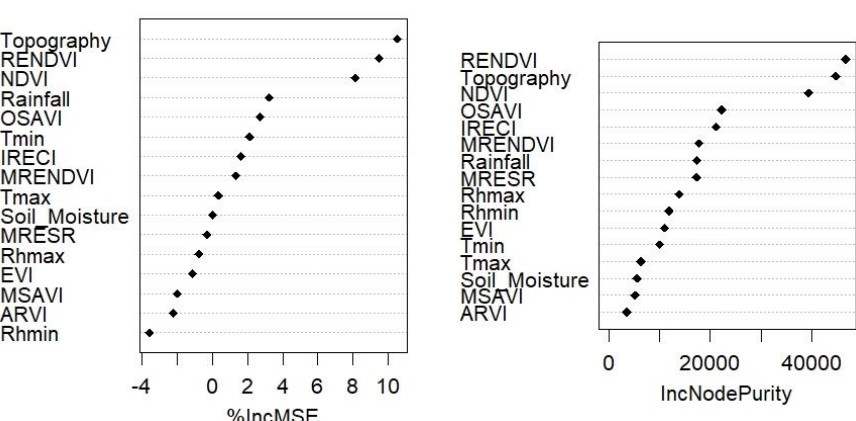

**Figure 3.** Variable importance plots for random forest regression model. Variable importance plots showing the relative importance of each variable as a predictor of aboveground biomass in the Cross River State, Nigeria. %IncMSE: increasing percentage mean square error. IncNodePurity: increasing node purity.

As shown in Figure 3, top parameters that made significant contributions to predicting AGB include topography, rainfall, NDVI, RENDVI, minimum yearly air temperature and

OSAVI. For instance, the elimination of topography and RENDVI as a predictive variable reduces the model performance to 55% compared to 85% when all the explanatory variables are included in the model. Conversely, variables, such as minimum relative humidity, ARVI, MSAVI, EVI, maximum relative humidity, MRESR, soil moisture and maximum yearly air temperature, may not have large effects on the model performance as shown in Figure 3. However, as revealed in Figure 3, considering both the %IncMSE and node purity, the important parameters exhibit instability in ranking.

In view of this, the top six parameters of the %incMSE were used to spatialize the AGB of the study area. The importance of these variables is further discussed in Section 4.1 of this paper. The application of these top six variables in AGB prediction saw a change in the model training accuracy yielding an $R^2$ of 0.78, an RMSE of 54.7 t/ha and an MAE of 34.89 t/ha compared to the training accuracy of the full predictors yielding an $R^2$ of 0.85 and an RMSE of 28.7 t/ha.

AGB from the testing forest inventory plots were used to determine the predictive accuracy of the final constrained RF model (Figure 4). The scatter plot of observed forest inventory AGB versus predicted RF AGB shows that the observed AGB aligned with predicted AGB with an $R^2$ of 0.88, an RMSE of 40.9 t/ha and a relRMSE of 29.96%. Separating this into 100 t/ha bins, AGB < 100 t/ha is predicted with an RMSE of 21.7 t/ha (66.5% relRMSE/10.1 t/ha bias), AGB between 100 and 200 t/ha is predicted with an RMSE of 47.5 t/ha (29.3% relRMSE/22.8 t/ha bias) and AGB >300 t/ha is predicted with an RMSE of 57.25 t/ha (18.5% relRMSE/−19.3 t/ha bias).

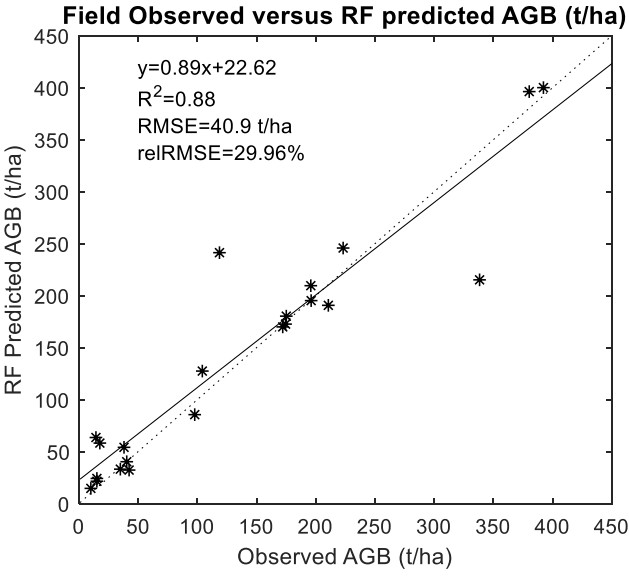

**Figure 4.** Evaluation of the random forest predicted AGB over the 22 testing forest inventory plots using the six most important predictor variables of %incMSE shown in Figure 2.

The spatial distribution of predicted AGB values and associated uncertainty over the Cross River State are presented in Figure 5. Over the Cross River State, high AGB is concentrated in two pockets: the south-eastern areas of the state (Oban area) and the north-eastern areas (Okwango area) coinciding with much of the CRS National Park. This area sees AGB above 200 t/ha and up to 500 t/ha. Areas around the Cross River to the south of the state, and scattered areas to the west of the state see AGB values of 150–350 t/ha. Areas to the far south, far west and north of the state have the lowest AGB below 100 t/ha. Average uncertainty over the CRS is estimated to be 34.6%, with lower percent uncertainty (0–50%) in higher biomass areas.

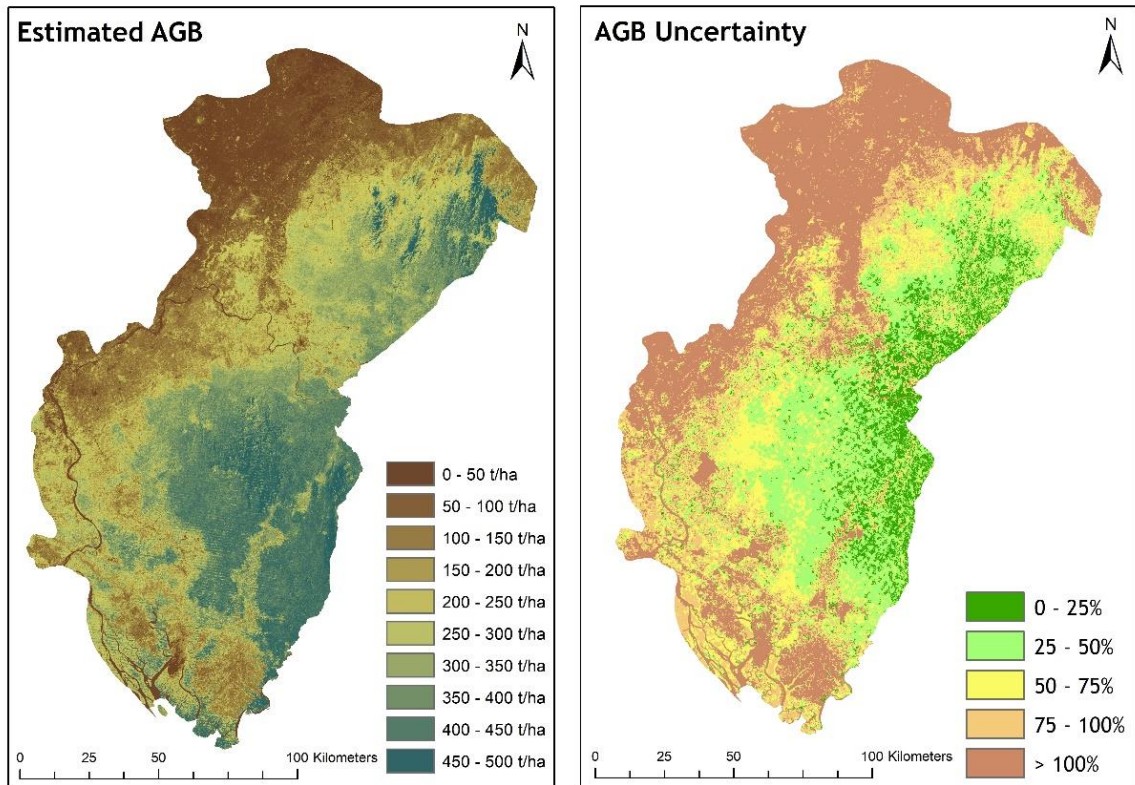

**Figure 5.** Estimated 20 m resolution map of aboveground biomass for the Cross River State (**left**-panel), with the resulting uncertainty in AGB incorporating prediction, measurement and allometry errors (**right**-panel).

*3.3. Comparison with Other Aboveground Biomass Products*

The Cross River State AGB product developed in this study was compared to the products from Saatchi, Baccini, Avitabile and ESA CCI+. We also included the REDD+ estimate of total AGB over the whole of Cross River State. Here, we compare distribution patterns, model performances of the four products as well as the mean, maximum and total AGB estimated over the Cross River State (Table 3). The average and total woody plot AGB estimated for the region in the current study is 121.98 t/ha and 0.246 Pg. Baccini's product is the closest to these results, with mean and total biomass at 86.87 t/ha and 0.253 Pg, respectively, within 29 and 3% of the current study's values, respectively. Saatchi's product has mean and total biomass values of 93.86 t/ha and 0.290 Pg, respectively, within 23 and 18% of the current study's values, respectively, and Avitabile's product has mean and total biomass values of 109.69 t/ha and 0.330 Pg, respectively, within 10 and 34% of the current study's values, respectively. The ESA CCI+ product is the most different with around 50% underestimated total biomass. Furthermore, the UN-Nigeria REDD+ estimate in 2018 also produced a close total biomass estimate of 0.267 Pg (8% larger than the current study's estimates). The distribution patterns of the regional estimates of AGB between the four products and the current study are given in Figure 5.

All the regional aboveground biomass products over the Cross River considered in this study are presented in Figure 6. The Saatchi map aligned with most regions of the AGB map of this study. The Saatchi product has similar magnitude AGB in the central (Figure 6, boxes B) and north-eastern highland areas, with high AGB values reaching 350–500 t/ha in these areas for both products. In addition, the Saatchi product contained low AGB along the western (Figure 6, boxes A), southern and north-western areas of the CRS with many of the predicted AGB values < 50 t/ha. The Baccini product has more consistent AGB to the current study in the western (Figure 6 boxes A), southern and north-western edge, but has

lower AGB in the central and north-eastern highland areas with AGB values from 250 to 350 t/ha (e.g., see boxes B).

**Table 3.** Mean, maximum and total AGB by products and study over Cross River State, Nigeria.

| Product/Study | Mean AGB t/ha | Maximum AGB t/ha. | Total AGB (Pg) |
|---|---|---|---|
| Saatchi et al. 2011 [22] | 93.86 | 365.9 | 0.290 |
| Baccini et al. 2012 [23] | 86.87 | 244 | 0.253 |
| Avitabile et al. 2016 [24] | 109.69 | 443.1 | 0.330 |
| UN-Nigeria REDD+ 2018 | - | - | 0.267 |
| ESA CCI+ 2021 | 71.71 | 205 | 0.124 |
| Current Study | 121.98 | 588 | 0.246 |

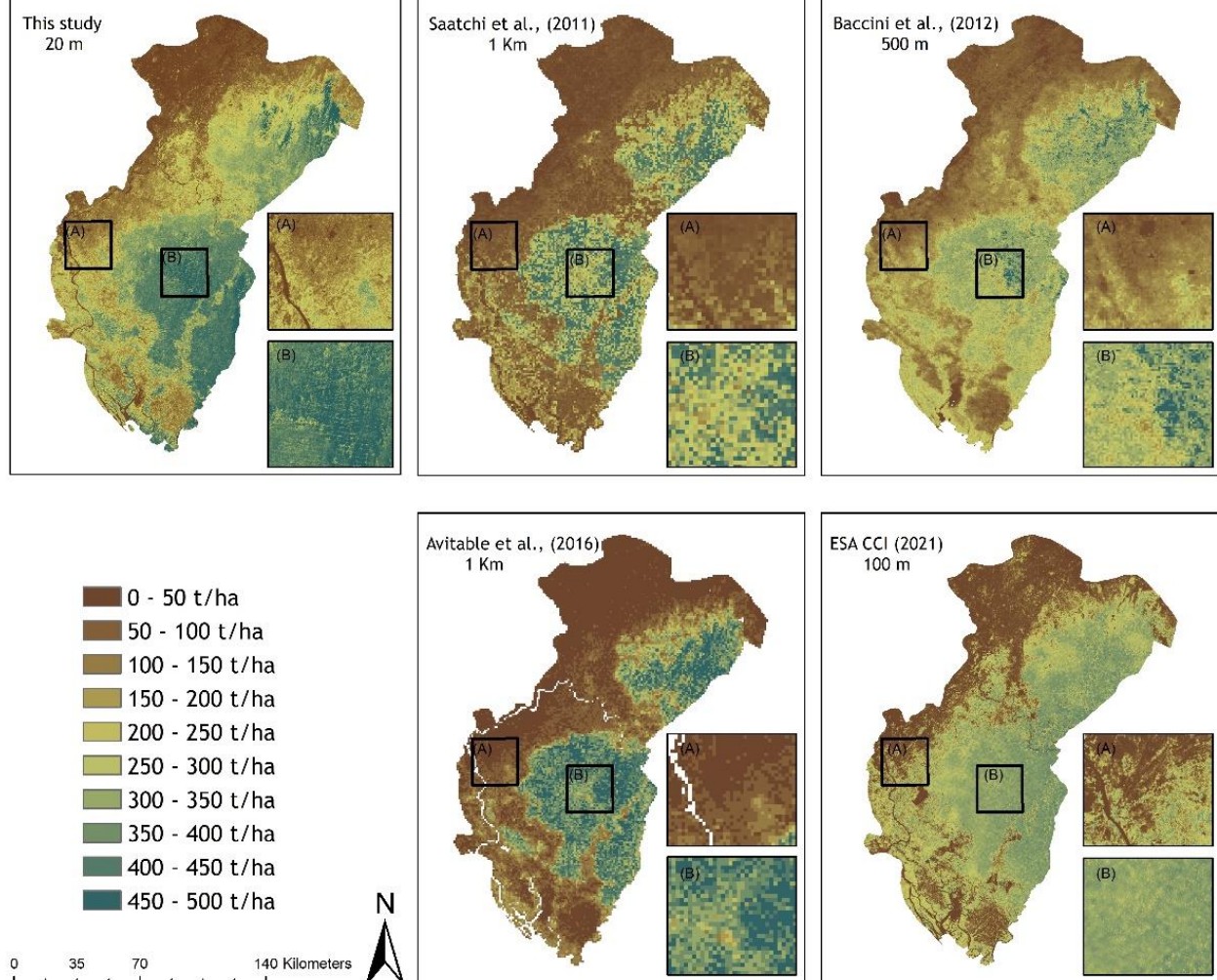

**Figure 6.** AGB products over the Cross River State of this current study and the Saatchi [22], Baccini [23], ESA CCI+ [25] and Avitabile [24] at their respective resolutions. Zoom in areas showing an area of transition from high to lower AGB areas (**A**) and high biomass areas in the center of the state (**B**) are also given. Boxes A and B indicate locations and so have the same name for all five AGB products.

The Avitabile product qualitatively compares most favorably to the AGB map of this current study in the central (Figure 6 boxes B) and northeastern highlands with

AGB values > 350 t/ha. Yet, as with Saatchi, southern, western and northwestern areas contained lower biomass values regularly below 50 t/ha. The current AGB map of the Cross River State by ESA CCI shows more homogenized AGB across much of the CRS. For instance, in the central (boxes B) and north-eastern parts of the study area, the ESA CCI product shows biomass only up to 350 t/ha with a gradual drop in AGB in the southern, western (boxes A) and north-western regions. Note that the dates of each product are 10 years apart (2011–2021) and thus biomass may be affected by anthropogenic and climate disturbances as well as natural ecological growth and mortality processes (see Discussion Section 4.2).

The performance of the four regional biomass maps is assessed against the 22 testing forest inventory plots with resulting metrics given in Table 4. The product that is closest to the observed forest inventory plots is the Avitabile product resulting in an RMSE of 32.89 t/ha and a relRMSE of 24.06%. The Saatchi AGB product contains errors of RMSE 67.62 t/ha with a relRMSE of 49.69%. The Baccini and ESA CCI products performed worse compared to others as they recorded an RMSE of 78.03 t/ha and a relRMSE of 57.09% and 78.87 and 56.24, respectively. These results are also confirmed using the similarity agreement index of Willmott, with the Saatchi and Baccini products yielded indices of 0.89 and 0.85, while the Avitabile and ESA CCI products yielding indices of 0.98 and 0.85 compared to the 0.97 obtained for this study. Concerning the bias and MAE, all products performed worse than the current study, with the Avitabile product being the closest (bias of −17.3 t/ha compared to +7.5 for the current study).

**Table 4.** Predictive mean errors of the AGB products of the Saatchi, Baccini, Avitabile and ESA CCI products over the Cross River State, Nigeria.

| AGB Product | RMSE (t/ha) | MAE | Bias (t/ha) | RelRMSE% | Willmott Index |
|---|---|---|---|---|---|
| Saatchi | 67.93 | 41.35 | −40.9 | 49.69 | 0.89 |
| Baccini | 78.03 | 48.41 | −48.4 | 57.09 | 0.85 |
| Avitabile | 32.89 | 23.57 | −17.3 | 24.06 | 0.98 |
| ESA CCI | 78.87 | 59.52 | −49.9 | 56.24 | 0.85 |
| This study | 40.95 | 23.14 | +7.5 | 29.95 | 0.97 |

## 4. Discussion

Nigeria, with over 200 million people and a land area of 923,768 km$^2$, has the highest rate of deforestation in Africa. According to Global Forest Watch [38], in the last 20 years Nigeria has lost 11,415 km$^2$ of tree cover, equivalent to 587 Mt of carbon dioxide emissions and 1530 km$^2$ of humid primary forest. Around 10% of this total tree cover loss and 23% of the primary forest loss has happened in the Cross River State. To halt this trend, Nigeria keyed into REDD+ in 2008 and formally received approval to kick start the project in the Cross River State in 2009. The decision to start with CRS was informed by the fact that 50% of the nation's remaining tracks of forest are found in the region, a valuable part of the Guinean forest biodiversity global hotspot [49]. The Paris agreement recognized forest protection as part of the strategy to counteract global carbon dioxide emissions, hence the need to quantify and track changes in biomass in forest and woodlands [16]. To achieve this, the IPCC [35] places emphasis on tier 3 level carbon accounting: reliance on local reference plots, tracking changes in activity data and institutionalization of monitoring, reporting and verification (MRV). However, existing efforts by the Cross River State REDD+ and other regional products fall short of internationally recognized standards due to a lack of local reference biomass data in both space and time in the region and across Africa, and a lack in consistency of data collection for monitoring and reporting on carbon dynamics at regional scales within the framework of REDD+ [101]. Subsequently, global, and regional attempts at carbon accounting [22–24] are characterized by large uncertainties attributed to this lack of, or inadequate, reference plots for the region. Coupled with the need for better forest

inventory reference data, new higher resolution remote sensing techniques such as Sentinel-2 and non-parametric machine learning methods can aid in reducing uncertainties in the prediction of tropical forest biomass pertinent to national carbon accounting, sustainable forest management, strategic policy making and REDD+ payment.

In view of these, the study aimed at deriving a high spatial resolution (20 m) above-ground biomass map for the year 2020 for the whole of the Cross River State, Nigeria, using Sentinel-2 data, climatic and edaphic variables and local reference forest inventory plots taken from undisturbed, disturbed and cropland areas. In addition, the constraining of predictor features in the random forest model helped in improving biomass prediction over the Cross River State while reducing predictor feature multicollinearity [90]. This study predicted spatially resolved AGB over the Cross River State of 0.246 Pg (average of 121.98 t/ha) with an RMSE of 40.9 t/ha, a bias of 7.5 t/ha, a relRMSE of 30% and an overall uncertainty averaged at 34.6%. REDD+ produced a single AGB estimate over the Cross River State of 0.268 Pg. The AGB prediction of this study is better compared to the regional products of Saatchi, Baccini and ESA CCI which yielded relRMSEs of 49.69%, 57.09% and 56.24%, respectively (bias of −41, −48, −50 t/ha), and similar to the Avitabile product (relRMSE of 24% and bias of −17 t/ha).

### 4.1. Aboveground Biomass Estimation over the Cross River State

Using all 16 predictor features including Sentinel-2 derived indices, climate variables and edaphic conditions resulted in a predicted AGB with a training RMSE of 28.7 t/ha and an $R^2$ of 0.85. Subsequently, the feature selection process was down to six features resulting in final training accuracy with an RMSE of 54.7 t/ha and an $R^2$ of 0.78. Of the 16 features, two climatic features were the most important: mean annual rainfall and minimum yearly air temperature and three Sentinel-2 derived indices were selected: NDVI, RENDVI and OSAVI and topography.

Topography exerted a very high influence on the distribution of AGB in the Cross River State (Figure 3), with higher AGB coinciding with areas of the CRS with higher topography. A principal reason for this is anthropogenic drivers of land cover change at lower elevations globally, but also around the Cross River State [13,39]. Deforestation and a history of agricultural use results in a loss of above ground biomass from agroforestry areas to larger-scale commercial cropland with limited tree cover. Most croplands identified by the CRS Forestry Commission are in the lower elevation areas of the state (Figure 1). Second, much of the upland areas of the CRS are occupied by forest reserves, such as the Cross River National Park separated into the Okwango (northeast) and Oban (southeast) sections consisting primarily of high biomass moist tropical forest. Third, topography itself can be a driver of higher biomass and biodiversity. Topography can shape climate regimes and influence diversification [102] as well as being linked to a range of abiotic conditions, such as soil water and nutrient availability, soil texture, exposure and flood regimes [7].

Rainfall and minimum yearly air temperature also exerted a strong influence on the distribution of AGB over the Cross River State. Climate heterogeneity is among the leading drivers of forest structure, biodiversity and aboveground biomass of tropical forest ecosystems [103,104]. Precipitation has a positive correlation with AGB [105,106] and over Africa has been estimated to be more important than other tropical continents due to lower average rainfall and larger water limitation over Africa [106]. Temperature has been shown to be negatively correlated to tropical forest AGB [105,106] with the temperature of the coldest month also negatively correlated with AGB [106]. Studies in western Africa, including Balima et al. [107] and Maukonen and Heiskanen [108] have also shown that within the west African region, mean annual rainfall from 800 to 1200 mm has a positive correlation with AGB and is negatively correlated with mean air temperature from 27 to 29 °C. Similarly, Poorter et al.'s [104] study revealed that lower air temperature supports soil fertility increase, and subsequently plant growth. Conversely, higher air temperature may reduce the rate of biomass growth. The Cross River State agroecological zones are characterized by varying climatic conditions [50]. The density of AGB across the

three ecological zones (north, south and central) possibly reflects the gradients of air temperature and precipitation conditions of the area. In the northeast and southeast flanks where rainfall often exceeds 2500 mm in most parts of the year, AGB is observed to reached 200 t/ha, whereas in northwest and southwest areas with less precipitation, AGB is generally below 150 t/ha (Figure 5, left panel).

Sentinel-2-derived NDVI, RENDVI and OSAVI were important predictors identified in this analysis to predict regional AGB over the study area (Figure 3). Specifically, the use of the red edge in the RENDVI has recently been shown to be effective in predicting forest AGB relaying issues with saturation at high biomass values and reducing uncertainties in complex and dense tropical forest [106,109]. Adan [110], for instance, compared the strength of red-edge and broad band-based VIs derived from Sentinel-2 in predicting total AGB in the tropical forest of Malaysia, concluding that the red-edge Vis, such as REDNVI, performed better than the non-red-edge VIs in predicting AGB. OSAVI was also used in this study to predict AGB. OSAVA is a known VI that enhances the contrast between soil and vegetation but aids in reducing the brightness effects of the soil [108].

*4.2. Comparison to Other Regional AGB Products*

As with this study, prior pan-tropical and global aboveground biomass products shown in Figure 6 have used a combination of satellite data and machine learning methods calibrated and validated using available forest inventory reference data. The total AGB predicted in this study over the CRS is closest to the Baccini and UN REDD+ estimates, and furthest away from the ESA CCI product (Table 3). Concerning the accuracy assessment (Table 4, this study performed better than the Saatchi, Baccini and ESA CCI products with around a 20–27% reduction in relRMSE and around a 27–38 t/ha reduction in RMSE. The Avitabile product has a similar but lower relRMSE (~6% better) but larger bias compared to our study (Table 4).

The Saatchi [22], Baccini [23] and Avitabile [24] products used the GLAS satellite sampling LiDAR (i.e., not wall-to-wall), calibrated using reference plots over the tropics to predict AGB, and then used MODIS multispectral data and satellite topography data to spatially extrapolate to the tropics using a machine learning algorithm. Avitabile is an improved product fusing Saatchi and Baccini using over 14,000 reference datasets (953 in Africa) to create a nearly unbiased product with a published mean bias of +5 t/ha and <+10 t/ha bias over Africa. The Avitabile product achieved prediction of higher AGBs in dense tropical forests >400 t/ha in Africa, around 100 t/ha more than the Baccini and Saatchi products [24] (see also Figure 6 and boxes B). Yet, Avitabile over the CRS still has an overall negative bias of −17 t/ha compared to our study with a +7.5 t/ha bias. The method developed here over the Cross River State has used localized forest inventory reference data collected explicitly for this purpose using the REDD+ Nigeria field team and spatially extrapolated using higher resolution multispectral Sentinel-2 data at 20 m as well as topography and climate data. Recent studies have begun to use Sentinel-2 to produce AGB maps for forests in Nepal (Pandit et al. [93]), Indonesia (Dube et al. [111]), Senegal (Soto-Navarro et al. [112]) amongst others. The ability in these Sentinel-2 studies, and the current study over the CRS, to predict AGB using various VIs outweighs the use of similar Landsat spatial resolution [113]. The ESA CCI+ biomass product included over 110,000 forest inventory reference plots from various global ecosystems and has largely used C and L-band radar data to determine global biomass [25]. Given the use of radar, the ESA CCI product begins to saturate at AGB values > 200 t/ha with a bias at 300 t/ha greater than −50 t/ha [25]. This study over the CRS predicts large AGB values (regionally >400 t/ha) with a bias from 200 to 400 t/ha at −19 t/ha and a relRMSE of 18.5%.

These biases and uncertainty within biomass products, emphasized the necessity for spatial extrapolation using field plots and remote sensing [88], and the uncertainties when comparing between products is likely responsible for the reluctance of the IPCC to recommend Saatchi et al. [22], Baccini et al. [23] and Avitabile et al. [24] biomass maps, hence its reliance on national forest inventories for subregional and regional biomass

spatialization [18]. However, all the regional products and our AGB map clearly identified similar areas—north-eastern and south-eastern flanks of the Cross River State—as areas with high biomass density. The differences observed in other locations of the study area and in the magnitude of the high AGB areas supports the need for better localized reference data [18] with higher resolution spatial imagery.

We do recognize that comparing our AGB map with these four products may incur errors associated with the differences in time scales between AGB products. In primary forests, AGB growth in tropical rainforests in Africa has been estimated at 0.5–2.1 Mg ha$^{-1}$ y$^{-1}$ [114]. In disturbed and secondary African tropical forests, AGB growth has been estimated to $-0.1$ to 5.5 Mg ha$^{-1}$ y$^{-1}$ [114] but can be counteracted by selective logging or other partial disturbances (see Section 2.2). Yet, this temporal issue cannot explain the large regional differences in the products, namely, the magnitude in AGB and sharpness of gradient of AGB decline west of the two high biomass regions in the central and northeastern CRS (see Figure 6 box inlets).

*4.3. REDD+ Implications and Future Work*

The leading mandates of REDD+ are to facilitate robust forest carbon quantification at different jurisdictional levels and maintain and improve on carbon status for carbon emissions reduction [28]. Because of this, nations are granted financial benefits based on their performances; judged on demonstrable evidence at slowing, halting or reversing forest cover destruction and carbon loss [28,115]. Therefore, the accurate estimation of aboveground biomass and mapping is pertinent, and this will reduce uncertainty in carbon stocks and cycle models especially in this part of the world where airborne LiDAR and field data remain a challenge [116]. The integration of AGB derived from satellite remote sensing and field measurements in our study increases confidence in our aboveground biomass estimation, unlike the UN-Nigeria REDD+ team, Saatchi, Baccini and ESA CCI products that either estimated AGB from field measurements alone or estimated AGB regionally with limited forest inventory plots over Nigeria. The method presented in this current study also does not rely on the improvement or fusing of prior products as with the Avitabile product. The disparity in the estimated AGB from these products has been linked to the different empirical modelling tools, calibration datasets and extrapolation techniques [25,88,116]. The IPCC Tier 3 Good Practice Guidebook emphasized accurate AGB reference data as vital in sustainable forest management and climate mitigation [37]. In addition, the accurate quantification of AGB is a requisite for meeting the four pillars of REDD+: National REDD+ strategy, national forest monitoring system and system of tracking REDD+ impacts on safeguards [117]. The Cancun Agreement outlined the social and environmental safeguards in Appendix 1 that implementing partners need to uphold [117]. Factual AGB estimation and monitoring of carbon stocks is one fundamental pathway to achieving this. In addition, with accurate AGB quantification in the region, land use policies will be put in place towards meeting the land degradation neutrality target 15:3:1 of UNEP/CBD/SBSTTA [118] and better the livelihood of forest dependent communities.

Future work may improve the AGB prediction of the CRS. First, the Global Ecosystem Dynamics Investigation (GEDI) satellite LiDAR has recently been attached to the International Space Station providing samples of forest structure globally. These LiDAR samples, coupled with on-the-ground biomass validation, could provide updated AGB maps with spatial extrapolation similar to Baccini and Saatchi. Furthermore, a new 1 km AGB product is being released by GEDI, which may provide improved estimates [119]. Second, Sentinel-1 radar could also be used to estimate biomass in isolation or using a fusion approach with Sentinel-2. Third, a better disaggregation of forest and land cover types over the region could have improved this work. These could include undisturbed tropical rainforests vs. dryer tropical forest and sparse forests, various managed plantation forests, mangroves, forest disturbance history and trees in non-forest environments, such as urban, agroforestry, pastures, etc.

This study faced challenges of adequate forest inventory plots. The cost of gathering data on trees limited the numbers of plots in this study to 72 despite the size of the study area. We recognize that a higher sample size may have improved the accuracy of the AGB estimates. In addition, management of forest communities' expectations was tricky; to achieve results, we remained upright with all community opinion leaders as previously similar exercises exaggerated the benefits of forest protection through promises of handouts from government.

**5. Conclusions**

Reduced uncertainty high-resolution carbon monitoring in tropical Africa across a range of woodland types is crucial to REDD+ improving carbon accounting, facilitating robust quantification at all jurisdictional scales and understanding areas of high biomass and biodiversity importance. The lack of reliable tree structure parameters for wall-to-wall aboveground biomass estimation and validation in Cross River State, Nigeria, as in other parts of the tropics, remain an immediate factor for high AGB uncertainty. In view of this, the study integrated in situ forest inventory plots collected over the whole state, and selected reanalysis climate data with Sentinel-2 derived vegetation indices to estimate regional aboveground carbon using random forest at 20 m resolution. The result revealed that Sentinel-2, climate variables and local forest inventories effectively predicted AGB over the whole of the Cross River State, Nigeria, with an RMSE of 40.9 t/ha, $R^2$ of 0.88, relRMSE of 30% and bias of +7.5 t/ha.

More so, the uncertainty and bias values obtained here, unlike the relatively high uncertainty of the Saatchi, Baccini and ESA CCI AGB products, reinforces Chave's et al. [18] call for the establishment of sampling plots across the tropics to improve biomass estimations. REDD+ in Nigeria provided only regional biomass rather than pixel-based spatially resolved biomass and used estimated tree height rather than the actual tree height measurement in the field. The AGB product derived from this study can serve as a baseline for REDD+ implementation, boost confidence in investment in tree carbon stocks, increase the conservation value of natural resources, reduce climate change impacts, and enhance the living standards of forest buffer communities.

**Supplementary Materials:** The following supporting information can be downloaded at: https://www.mdpi.com/article/10.3390/rs14225741/s1, Supplementary Materials S1—Photographs of Forest Inventories, S2- AGB map over Cross River State, Nigeria

**Author Contributions:** U.A.A. with doctoral supervision from A.S.A. and Y.W., conceptualized the study, led the field work team, analyzed the data and wrote the paper. Technical inputs were offered by B.F.E. and C.J.I. All authors have read and agreed to the published version of the manuscript.

**Funding:** Funding of this research was made possible by the Tertiary Education Trust Fund of the Federal Government of Nigeria through the Federal College of Education (FCE) Obudu.

**Data Availability Statement:** The aboveground biomass regional product derived over the Cross River and other data presented in this study are available on request from the corresponding author.

**Acknowledgments:** Funding of this research was made possible by the Tertiary Education Trust Fund of the Federal Government of Nigeria through the Federal College of Education (FCE) Obudu. We cherish this support. We also use this opportunity to acknowledge the support of all the community liaisons without whom it might have been difficult to overcome the challenges we faced in the field.

**Conflicts of Interest:** The authors declare no conflict of interest.

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
