# Peer review of "Quantification of Above-Ground Biomass over the Cross-River State, Nigeria, Using Sentinel-2 Data"

_remotesensing, doi:10.3390/rs14225741_

Round 1

Reviewer 1 Report (New Reviewer)

Author Response

Reviewer 2 Report (New Reviewer)

Dear Authors,

A very good paper, well set out and I found it interesting.

I do, however, have a couple of points that I would like to have clarified.

a) the classification is for a single year. How sensitive is the classification to season? In my experience (in Sierra Leone) is that there is a relatively short period in the dry season when good cloud free, dust free and smoke free images can be obtained. Even within this short period of time the "appearance" of the vegetation can change a lot - this is especially noticeable in woodlands and farm bush (less so in forests). So the question is, "if you repeated this exercise next year could you distinguish real (on the ground) change from 'error' (in the sense of uncertainty and of different images)?". That is are you confident that if this exercise were repeated the "signal to noise" ratio would be high enough to be meaningful?b) I would like to see some examples of where there are differences between the different data sets in more detail. Figure 6 is good and should be retained but one or more areas of say 1 or 2 sq km would allow a clearer understanding of where the difference lie.

b) Figure 6 is good but I would like that in addition you "zoom in" to 2 or 3 small areas (1 or 2 sq km) so that it is easier to see where the differences are?

c) I would quite like to see pictures of, say 3, of the ground plots to indicate how the vegetation looks for different AGB (high medium and low).

Author Response

Reviewer 3 Report (Previous Reviewer 2)

The authors have incorporated the proposed modification comments, which makes the manuscript has a great improvement. Now, this version is suitable for publication in the Remote Sensing. Some minor details should be revised:

Line 316 Equation (2)->(2)

Line 317 Equation (3)->(3)

Line 324 Equation (4)->(4)

Line 346 Equation (5)->(5)

Please delete title "AGB" of Figure 3

Author Response

This manuscript is a resubmission of an earlier submission. The following is a list of the peer review reports and author responses from that submission.

Round 1

Reviewer 1 Report

This paper addresses an important and interesting problem-quantification of above-ground biomass using Sentinel-2 data. The authors use Sentinel-2 multispectral imagery combined with climatic and edaphic variables to estimate the regional distribution of above-ground biomass (AGB) over the Cross River State, a tropical forest region in Nigeria, using the Random Forest (RF) machine learning.

Overall, this article is well organized and its presentation is good. However, some issues still need to be improved.

(1)   Line 182: what is the location accuracy and type of GPS?

(2)   Line 209: Add a flowchart in Section 2.4 to make the structure of this manuscript clear to readers as a suggestion.

(3)   Line 209: in this section, authors cannot tell readers how to use Sentinel-2 images with investigation data to map AGB products, including original data, processed data, the implementation process and so on. Please give a detailed statement to describe the whole process or flowchart to generate AGB products.

(4)   Line 215: where were Sentinel 2A images downloaded? Please insert download link here.

(5)   Line 223: The spatial resolution of VIS and NIR (10 m) and red-edge and SWIR (20 m) of Sentinel 2 image. How did you make the spatial resolution of different bands to be consistent? Please give an explanation here.

(6)   Line 235: How many images were used in this study? Please insert the total number of images used or give a detailed list in the manuscript if possible.

(7)   Line 245: Give an explanation of default extraction procedure if no explanation, readers cannot understand it. Another question is: the spatial resolution of Sentinel 2 is 20 m. How do you make that the center point of the plot is just right in the middle of a pixel of Sentinel 2?

(8)   Line 267: The accuracy of AGB estimation using RF mainly depends on the quantity and quality of sample points. How do you guarantee or solve these two issues?

(9)   Line 448: 55% and 85%, how did you get these two number? Have a test or get it in a literature?

(10) Line 519: why the Baccini AGB map seems to tally with the predicted AGB of this study in areas with low AGB? Could you give a reason to explain this phenomenon?

(11) Line 522: AGB products from the Saatchi, Baccini, ESA CCI+ and Avitabile have different spatial resolution. Does different spatial resolution have an influence on result analysis?

(12) Line 527: Is it possible that methodology difference leads to AGB difference between this study and the Avitabile product? Give relative analysis about this issue.

(13) Line 631: add the discussion on the effectiveness and feasibilities of this study as a suggestion.

(14) Line 656: what is the advantages of this study compared with other methods or products?  

Reviewer 2 Report

This manuscript will provide scientific contributions to the studies on wall-to-wall aboveground biomass estimation in tropical Africa across a range of woodland types. There is a lot of great work in this manuscript. The manuscript is logically structured and reads well. The description of the objectives, methods and results is mostly clear and comprehensible. The highlights are also clarified clearly in the Introduction. I suggest this manuscript need to be revised and improved (see comments below).

Detailed comments:

1.       A few of typos, spelling mistakes etc. in the paper. So I recommended the author to carefully read through the draft to correct each typo.

2.       Please add names of manufacturer or company for software used in this study, such Line 232, 245.

3.       The Section 3.1 describes the field data results. I suggest that the Section 3.1 should be placed at the Section 2.2.

4.       Line 69-71: the authors used three different symbols of the measurement unit, i.e., “t.ha.”, “t ha-1”, “t/ha”. Please use a unified writing method.

5.       Line 62, 66, 371, 390, 674: Same question as in the comment 4. “Lidar”, “lidar”, “LiDAR”.

6.       Line 452, 458, 465: table 3-> Table 3, figure 2-> Figure 2.

7.       Why did you develop area-specific (e.g., undisturbed and disturbed) models for estimating AGB? It is recommended to develop area-specific models with the land cover classification data and compare their results with the results of non-specific predictive models.

8.       Line 249, 251: “30 m”, “20m”, please add a space character between numbers and units through the whole manuscript. e.g., “20m”-> “20 m”.

9.       Line 309: “studies (83)”-> “studies [83]”.

10.   Line 509: “Table: mean and total AGB”-> “Mean, maximum and total AGB.

11.   Too many references!

12.   I think that too many variables input into the predictive models. I suggest that the authors should constrain the number of candidate variables entering the RF models after conducting a feature selection.

13.   Line 710: “R2 0.88”-> “R2 of 0.88”.

Reviewer 3 Report

Reviewer’s Report on the manuscript entitled:

Quantification of Above-Ground Biomass Over the Cross-River State, Nigeria Using Sentinel 2 Data

The authors applied Random Forest to Sentinel-2 imagery in combination with climatic and edaphic variables to estimate the regional distribution of above-ground biomass (AGB) over the Cross River State in Nigeria. Although the topic and results are interesting, the presentation should be improved. Below, please see my comments.

Line 48. Please define AGB. Please note that all the abbreviations must be defined the first time they appear (in addition to Abstract). Please also add an acronym table at the end of the manuscript.

Lines 97, 106, etc. Please remove “see”

Line 102-103. Please avoid using italic font.

Line 177. Is (51) a reference? Please use [].

Line 212. It should be “Sentinel” with capital S.

The structure is a bit confusing. Typically, it is recommended to have

Section 2.1 Study region

Section 2.2 Datasets and pre-processing

Section 2.3 Methods: In here please also show a flowchart of the methods/processes.

Your current section 2.2 should be in the method section.

Line 239. Please add the following recent article that describes and compares EVI and NDVI when it comes to monitoring and change detection:

https://doi.org/10.3390/rs12152446

Also, for SAVI and ARVI, etc. please add this

https://doi.org/10.3390/s21062115

And this review https://doi.org/10.3390/agriculture11050457

Equations and tables look like they are images. They should be typed in instead.

Figure 1. Please use a consistent format for all the texts and numbers in the figure. Please also enlarge the font size. The latitudes and longitudes and legend should be enlarged.

Figure 3. Where are the data points? Please plot them so the regression line and R^2 will make sense.

In Table 1, please use parentheses around the equations as they are currently incorrect. For example, say NIR-R/NIR+R should be (NIR-R)/(NIR+R) also remove NDVI right before it. Similarly for other equations.

Table 2. Please remove the dot after Max and Min.

Lines 608-609. The first article that I suggested above also describes a time-frequency analysis which shows how the climate and vegetation are coherent and can present time delay between their seasonal cycles. I suggest discuss this a bit here.  

Line 884. This should be [66] not [68].

In the Conclusions, please mention the limitations of the study and future direction.

Thank you for your contribution

Regards,

Round 2

Reviewer 1 Report

In general, this manuscript was well rerivesd and was qulified to RS.

Reviewer 3 Report

Dear authors,

Thank you for preparing the revisions. I had a careful look to the results of you manuscript and based on your responses, I am afraid that I cannot recommend acceptance. The reasons are as follows:

The title mentioned Sentinel-2 and I only see that you used 8 imagery (line 225) for quantifying the above ground biomass that is already a red flag because the temporal changes cannot be analysed using a few images and the results will be biased of seasonal change, natural disasters, atmospheric errors, etc., regardless of what method you used. There are many satellite imagery available that can be accessed through US Geological Survey or Google Earth Engine that can be used for AGB quantification. 

Also, I do not see the point clouds in your new Figure 4.

Round 3

Reviewer 3 Report

The authors tried to quantify above-ground bio-mass over the cross river state of Nigeria by using only a limited number of Sentinel 2 data. I found the manuscript poorly written, and the results are superficial. I do not see any justification of using only these limited satellite data sets. There are many freely available satellite data in US Geology Survey or Google Earth Engine (EEG) that authors can utilize for such analysis. The few Sentinel 2 images used here cannot capture the spatiotemporal  dynamics of the region, and so the results do not reflect the bio-mass distribution over the study region, therefore, I cannot recommend this article for Remote Sensing.